# Quality of life and associated factors among Norwegian public health nurse students during the first months of the COVID-19 pandemic: A cross-sectional study

Lisbeth Valla[1,2]*, Victoria Telle Hjellset[1], Milada Cvancarova Småstuen[1], Bente Sparboe-Nilsen[1]

1 Department of Nursing and Health Promotion, Faculty of Health Sciences, Oslo Metropolitan University, Oslo, Norway, 2 Regional Centre for Child and Adolescent Mental Health, Eastern and Southern Norway (RBUP), Oslo, Norway

* lisval@oslomet.no

**Data Availability Statement:** All relevant data are within the paper

## Abstract

The main objective of this study was to investigate the association between quality of life (QOL) and civil status, self-efficacy, loneliness, and physical and mental health among public health nurse (PHN) students during the COVID-19 pandemic in April 2020. PHN students were recruited from eight universities and university colleges in Norway. A range of potential predictive factors were entered into a model using a stepwise linear regression approach. In general, the PHN students reported a high level of QOL during the first month of the pandemic. The students' civil status, perceived physical health, self-efficacy and loneliness were all significantly correlated with QOL. Among these, the strongest predictive factors for QOL were found to be perceived loneliness and self-efficacy. Our results provide insight into the QOL of PHN students, which managers in the higher education sector can use to outline specific coping strategies that can help students during a pandemic.

## Introduction

The coronavirus disease 2019 (COVID-19) and the subsequent societal response to mitigate its effects had a disruptive impact on many students. including Norwegian public health nurse students (PHN). The World Health Organization (WHO) declared COVID-19 a pandemic and recommended social distancing guidelines to slow the spread of the virus [1]. The implementation of social distancing led to unexpected changes for many students, including inhibiting their educational activities. Based on rising concern about the pandemic, all universities in Norway postponed or cancelled all campus activities and modified the content of their various courses and programmes from face-to-face to online teaching activities from March 2020. This resulted in the students missing out on social day-to-day activities with the other students and teachers, in addition to experiencing general worry and uncertainty about the effect of the global pandemic. This was also the situation for Norwegian PHN students doing post-graduate studies in health promotion and sickness

**Funding:** The author(s) received no specific funding for this work.

**Competing interests:** The authors have declared that no competing interests exist.

prevention work for parents and their children between the ages of 0 and 20. Additionally, their ten-week training phase was also delayed or modified. Although the restriction of social contact was fundamental to reducing the spread of the virus, the psychological cost of the COVID-19 pandemic may have been high for many students [2, 3], which in turn posed a risk to their quality of life (QOL) [4, 5]. Inadequate efforts to recognize and address University students' challenges, especially during a pandemic, could have long-term consequences on their health and education [6]. We thus need more research on the factors that impact students' QOL to ensure that they are followed on. QOL is a multidimensional construct that theorists have conceptualised in multiple ways, including satisfaction with life [7], subjective well-being [8] and happiness [9]. In the present study, we use the term QOL to characterise a broad range of factors and define overall quality of life in accordance with Diener (1984) as subjective well-being. This is related to an evaluation of how people feel and think about their lives, which will here be referred to as appropriate coping strategies [10, 11].

There are several factors that can explain QOL in university students, including many of the following: gender, education environment, years of study, depression, chronic illness [12–14] and physical activity [15]. Also, poor social support and negative psychosocial factors such as loneliness [16] are associated with lower QOL, while positive psychosocial factors, such as high self-efficacy, have a positive impact on well-being and QOL outcomes [17]. Self-efficacy includes individuals' thoughts concerning their successes and failures, and their perceptions of feedback they have received [18]. Previous studies have shown that a high degree of self-efficacy is related to higher self-belief. Self-efficacy is also one of the most important factors contributing to behaviour change and may have a buffer effect on negative psychosocial factors such as stress in students [19, 20]. However, there is great variation in how well individuals tolerate the same load and how well they cope with new situations. Coping is a way of responding to a threat by action and, among other things, depends on previous experience and social support [21]. Lack of coping may be associated with psychological distress, and poor coping strategies may be a vulnerability factor in stressful situations. In contrast, high self-efficacy can buffer stressful environments [22] and influence the relationship between stressors and mental health outcomes.

Few other countries offer similar educational opportunities in health promotion and illness prevention work, although our graduates can be compared to students in other countries at a post-graduate or higher University level. Thus, this study can add to the existing pandemic literature in several ways. First, to the best of our knowledge data regarding psychological complications, feelings of loneliness and QOL among university students during the COVID-19 pandemic are scarce, and little research has also been done on protective factors such as self-efficacy in university postgraduate samples. Our study closes this information gap by identifying key factors that influence QOL in PHN students. From a health promotion perspective, more knowledge is needed about the determinants of students' QOL. The WHO emphasises QOL as a goal for public health and underlines the need for research that identifies key determinants for well-being and health in young people. Given the potential value of understanding what impacts university students' QOL to develop intervention programs and targeted learning activities, this study will be especially interesting to those who work with students, in addition to authorities and health policies that target university students.

In this study we aimed to examine the associations between QOL and civil status, self-efficacy, loneliness, and physical and mental health in PHN students in Norway in the context of the first wave of the COVID-19 pandemic.

## Methods

### Sample and data collection

This cross-sectional study includes PHN students from eight universities and colleges in Norway and covers all PHN study programmes in Norway. The higher education institutions vary in size and location (from city to suburb) and the programmes include students with different sociocultural background. The participants were recruited through the faculties' learning management system (LMS) where a link to the electronic survey was shared. All students, both full-time and part-time, enrolled in PHN educational programme in eight Universities and Colleges in Norway during the COVID-19 pandemic met the inclusion criteria for this study. The participating universities were Oslo Metropolitan University, VID Specialized University, University of South-Eastern Norway (USN), the Norwegian University of Science and Technology (NTNU), University of Stavanger (UIS), The Arctic University of Norway (UIT), Inland Norway University of Applied Sciences and The Western Norway University of applied Sciences. Data was collected through an anonymous web survey 'Nettskjema' [23], and was completed from May to November 2020 (2–8 months into the Norwegian lockdown). The survey took 20 minutes to complete, and a total of 275 students participated (response rate 63%).

### Measures

**Demographic variables.** The first part of the questionnaire comprised self-reported data on demographics. The following sociodemographic variables were collected: university/university college, age, gender, civil status, and children (yes/no).

**Quality of life.** Quality of life was measured using the Satisfaction with Life Scale (SWLS) (10). The SWLS includes five items. The items are 1, in most ways my life is close to my idea. 2, the conditions of my life are excellent. 3, I am satisfied with my life 4, so far I have gotten the important things I want in life. 5, if I could live my life over, I would change almost nothing. The scales are rated on a Likert scale ranging from 1 = Strongly Disagree to 7 = Strongly Agree. The mean scores from the five items were computed. The scale revealed a satisfactory internal consistency, with an α value of 0.89. Several studies have confirmed the validity of the SWLS [10, 24]. The answers were recoded so that higher values always indicated greater well-being. Studies comparing the SWLS in student populations in 42 countries have shown good psychometric properties [25].

**Self-efficacy.** Self-efficacy was measured using the Norwegian version of the Generalized Self-Efficacy Scale (GSE), which measures optimistic self-beliefs in coping with the demands, tasks and challenges of life in general [26]. The GSE consists of 10 statements, 1- I can always manage to solve difficult problems if I try hard enough, 2- if someone opposes me, I can find means and way to get what I want, 3- it is easy to me to stick to my aims and accomplish my goals, 4- I am confident that I could deal efficiently with unexpected events, 5- thanks to my resourcefulness, I know how to handle unforeseen situation, 6- I can solve most problem if I invest necessary effort, 7- I can remain calm when facing difficulties because I can remain on my coping abilities, 8- when I am confronting with a problem, I can usually find several solutions, 9- if I am in trouble I can usually think of something to do, and 10- no matters what comes my way, I am usually able to handle it. The respondent rates on a scale range from 1 (completely wrong) to 4 (completely right). The respondent's scores on each item are summed and divided by ten to achieve a GSE score ranging from 1–4, with higher scores indicating higher levels of generalised self-efficacy. The GSE has been shown to be reliable and valid [27].

**Mental health.** The students' mental health was assessed using the question, 'How satisfied have you been with your mental health, emotions and mood over the last four weeks?'

[28]. The respondent rates are given on a scale from 1 (very dissatisfied) to 10 (very satisfied). The mean scores of the ten items were computed and higher mean scores indicated a higher level of mental health. The mental health question has been tested and validated in a Norwegian setting [29].

**Physical health.** The students' physical health was assessed using the question 'How satisfied have you been with your physical health over the last four weeks?' [28]. The respondent rates are given on a scale from 1 (very dissatisfied) to 10 (very satisfied). The mean scores of the ten items were computed and higher mean scores indicated a higher level of physical health. The physical health question has been tested and validated in a Norwegian setting [29].

**Ethical considerations.** The data was collected through an anonymous web survey. The current study was approved by The Regional Committees for Medical and Health Research Ethics (2020/143629/REK-Sør-Øst) and the Norwegian Centre for Research Data (NSD). Written informed consent was obtained from all participants upon recruitment and participants did not receive any financial compensation Since the study is anonymous and Identifiable or sensitive data were not collected, REK concluded that there was no need for full ethical evaluation. To ensure full anonymity, we used broad categories for sociodemographic variables. There are no conflicts of interest involved in the project and no external funding.

## Statistical analysis

Continuous data are described by mean and standard deviation (SD) and categorical data by counts and percentages. Possible associations between selected predictive variables and QOL were modelled using multiple linear regression. All of the potential predictive factors were entered into the model using a stepwise linear regression approach. The results are expressed as regression coefficients (B) with 95% confidence intervals (CI) and effect sizes (ES). In line with Cohen (1962), effect sizes are classified as small ($d = 0.2$), medium ($d = 0.5$) and large ($d \geq 0.8$). The model fit was good and all assumptions for linear regression were fulfilled. To check and correct for possible biases, we constructed all confidence intervals using the bias corrected accelerated method (Bca). All tests were two-sided and p-values $<0.05$ were considered statistically significant. The analyses were performed using SPSS version 27.

## Results

Table 1 shows the sample characteristics. In total, 247 participants (97.8% females), most of whom were between 30–40 years of age (53.8%), were included in the analysis. In total, more than two-thirds of the participants lived with a partner and had children (Table 1). Participants were generally satisfied with their status of life satisfaction, with a median score of 22, a mean of 10 and a max score of 25. However, about 30.8% of the students stated that they felt lonely sometimes or very often, and 19.8% stated that they were not satisfied with their mental health. The median and range of QOL, satisfaction with mental and physical health and loneliness scores were 4.4 (range = 2–5), 7 (range = 2–10), 7 (range = 1–10) and 4 (range = 1–5), respectively.

Table 2 shows descriptive statistics for QOLcivil status, self-efficacy, loneliness, and physical and mental health.

Table 3 shows the association between QOL and civil status, self-efficacy, loneliness, and mental and physical health.

To assess the impact of the selected possible predictive factors, we applied a multivariate linear regression with the following covariates: age, gender, self-efficacy, satisfaction with mental health, satisfaction with physical health, loneliness and civil status. In our data, age and gender were not found to be associated with QOL.

**Table 1. Study population.**

| Gender (N = 275) | n (%) |
|---|---|
| Female | 269 (97.8) |
| Male | 6 (2.2) |
| Age (N = 275) | |
| < 30 years | 57 (20.7) |
| 30–40 years | 148 (53.8) |
| >40 years | 70 (25.5) |
| Civil status (N = 275) | |
| Married/cohabiting | 236 (85.8) |
| Divorced/single | 39 (14.2) |
| Children (N = 275) | |
| Yes | 214 (77.8) |
| No | 61 (22.2) |

In the multivariate analysis, our data revealed that the strongest predictive factors for QOL were self-efficacy (ES = 0.31) and loneliness (ES = 0.24). In addition, satisfaction with physical health, civil status and loneliness were all significantly correlated with QOL. Participants who were not married/living with a partner had QOL scores of around 0.17 points lower than participants living with a partner. The R-square was 0.554 and the adjusted R-square was 0.546 thus the included covariates explained about half of the variation in the dependent variable.

## Discussion

This study aimed to assess possible associations between QOL and civil status, self-efficacy, loneliness, and physical and mental health in Norwegian PHN students in the first wave of the COVID-19 pandemic. We found that the PHN students generally reported a moderate to high levels of QOL, which is comparable to QOL in a nationwide survey of Norwegian students in the non-pandemic period where Norwegian students in higher education generally scored moderate to high on the SWLS Scale [30, 31], However, civil status, physical health, self-efficacy, and a feeling of loneliness were all significantly correlated with QOL, but the strongest predictive factor for QOL among Norwegian PHN students was their perceived feeling of loneliness and self-efficacy. These findings are in line with previous research emphasising that self-efficacy [17, 20, 32] and perceived feelings of loneliness [2, 16, 33] are important predictors of QOL. Kuczynski et al. showed among undergraduate students (n = 1456) from four universities across the United States that interpersonal factors in general and loneliness were strongly associated with QOL [16]. Moreover, Saleh et al. demonstrated among 483 French college students aged 18 to 24 years that self-efficacy was the most important predictor of stress. At the same time Diotaiuti et al. showed among 707 university students from Italy that self-efficacy is substantially connected with QOL [17]. It is important to note that the participants in the current research are older than those in earlier studies and are enrolled in postgraduate programs; they have spent several years at the University completing these degrees. However, our findings among post graduate students support earlier results that self-efficacy and loneliness are connected to QOL.

This finding is of great interest, especially considering that social distancing and the lockdown served a bigger purpose to stop the spread of the virus and, thus, the perception of loneliness may have been part of the required adaptation to the circumstances. However, guidelines suggested avoiding isolation and keeping in touch through technology [1], and studies have found that online interactions can foster a sense of connection, enhance well-being and thus

**Table 2. Descriptive statistics and scores for selected analysed variables (QOL, civil status, self-efficacy, loneliness, and physical and mental health).**

| Variables | Descriptive statistics | | | | |
|---|---|---|---|---|---|
| | median | Range (min, max) | | | |
| Physical health | 7 | 1–10 | | | |
| Mental health | 7 | 2–10 | | | |
| | Scores | | | | |
| | 1 | 2 | 3 | 4 | 5 |
| | n (%) | n (%) | n (%) | n (%) | n (%) |
| Loneliness | 5(11.8) | 20(7.3) | 81(29.5) | 123(44.7) | 46(16.7) |
| Civil Status | 131(47.6) | 105(38.2) | 8(2.9) | 31(11.3) | |
| **Satisfaction with Life Scale (SWLS)** | | | | | |
| In most ways my life is close to my ideal. | 6(2.2) | 25(9.1) | 46(14.5) | 142(51.6) | 62(22.5) |
| The conditions of my life are excellent. | 1(0.4) | 7(2.5) | 13(4.7) | 104(37.8) | 150(54.5) |
| I am satisfied with my life | 0(0.0) | 6(2.2) | 15(5.5) | 92(33.5) | 162(58.9) |
| So far, I have gotten the important things I want in life. | 1(0.4) | 11(4.0) | 19(6.9) | 107(38.9) | 137(49.8 |
| If I could live my life over, I would change almost nothing | 11(4.0) | 19(6.9) | 64(23.3) | 104(37.8) | 77(28.0) |
| **Self-efficacy (GSE)** | | | | | |
| I can always manage to solve difficult problems if I try hard enough | 3(1.1) | 8 (2.9) | 169(61.5) | 95(34.5) | |
| If someone opposes me, I can find means and way to get what I want | 6(2.2) | 55(20.0) | 186(65.5) | 34(12.4) | |
| It is easy to me to stick to my aims and accomplish my goals | 2(0.7) | 20(7.3) | 196(71.3) | 57(20.7) | |
| I am confident that I could deal efficiently with unexpected events | 1(0.4) | 22(8.0) | 190(69.1) | 62(22.5) | |
| Thanks to my resourcefulness, I know how to handle unforeseen situation | 1(0.4) | 20(7.3) | 182(66.2) | 72(26.2) | |
| I can solve most problem if I invest necessary effort | 0 (0.0) | 14(5.1) | 163(59.3) | 98(35.6) | |
| I can remain calm when facing difficulties because I can remain on my coping abilities, | 3(1.1) | 39(14.2) | 172(62.5) | 61(22.2 | |
| When I am confronting with a problem, I can usually find several solutions | 1(0.4) | 24(8.7) | 196(71.3) | 54(19.6) | |
| If I am in trouble, I can usually think of something to do | 0(0.0) | 8(2.9) | 184(66.9) | 83(30.2) | |
| No matters what comes my way, I am usually able to handle it. | 0(0.0) | 24(8.7) | 173(62.9) | 78(28.4) | |

* Physical health: rated on a scale from 1 (very dissatisfied) to 10 (very satisfied).

* Mental health: rated on a scale from 1 (very dissatisfied) to 10 (very satisfied).

* Loneliness: rated on a scale from 1(very often) to 5(never)

*Civil status: 1 = married, 2 = Cohibitant,3 = divorced, 4, = singel

* Satisfaction with Life Scale (SWLS): rated a scale range from 1 = Strongly Disagree to 5 = Strongly Agree.

* Self-efficacy (GSE): rated on a scale range from 1 (completely wrong) to 4 (completely right).

reduce feelings of loneliness [34]. Social distance is as such replaced by physical distance, implying that there is still a social relationship between individuals, even if they are physically separated [35]. The Norwegian PHN students continued their studies from home and were

**Table 3. Association between QOL and civil status, self-efficacy, loneliness, and mental and physical health examined by multivariate linear regression analysis.**

| Variable | B | 95% CI | ES | p-value |
|---|---|---|---|---|
| Self-efficacy | 0.51 | 0.38; 0.65 | 0.31 | <0.001 |
| Satisfaction with mental health | 0.04 | -0.01; 0.09 | 0.12 | 0.057 |
| Satisfaction with physical health | 0.04 | 0.01; 0.07 | 0.13 | 0.003 |
| Loneliness | 0.19 | 0.12; 0.25 | 0.24 | <0.001 |
| Civil status (ref = married/cohabiting) | -0.17 | -0.22; -0.17 | -0.23 | 0.003 |
| Age | -0.90 | -0.17; -0.10 | -0.88 | 0.028 |

*Adjusted for age

able to interact with others through digital classes and groups. Despite these opportunities, however, it seems that some students still experienced feelings of loneliness, which adversely affected their QOL. The direct association between loneliness and QOL is consistent with prior research suggesting that social distancing and perceived loneliness increase stress and reduce well-being among undergraduate students in Switzerland during the COVID-19 pandemic [2].

The pandemic was a stressful situation for many of the students, and their QOL should therefore be considered in a bio-phyco-social context. Stress is a state of the mind involving both brain and body as well as their interactions [36]. However, how and to what degree major life events and the pressures of daily life affect individuals and alter their physiological systems differs, and there is also variation in how well students tolerate the same load and cope with new situations. Poor coping strategies may be a vulnerability factor in stressful situations. Our results showed that PHN students with higher self-efficacy had a higher QOL.

This may indicate that a strong belief in their abilities and appropriate coping strategies may buffer stressful environments [22], thereby having a direct, positive association with QOL, as individuals who scored higher on these variables reported a higher QOL. People who have high self-efficacy might see potentially challenging situations as opportunities rather than risks compared to people with poor self-efficacy, who are more likely to use highly adaptive coping mechanisms [37]. Students with a high level of self-efficacy may perform more challenging tasks, and this quality is therefore of great importance in school settings, particularly in extraordinary situations such as the social lockdown resulting from the pandemic presented here.

Since previous research has also documented that university and university college students experienced a variety of psychological problems during the COVID-19 pandemic [3], more attention should be paid to the outcomes of the pandemic in this group. QOL constitutes an important goal for individuals and societies with positive consequences for both individual and population health [38, 39]. It is therefore essential to be aware of risk factors relating to lockdown and similar situations. Determinants of QOL may provide valuable information that can inform the development of intervention programmes and targeted learning activities, thereby having important implications in educational practice. Therefore, the implications of this study are an important contribution for health policies that target university students in Norway, as most universities did not have the choice to not limit face-to-face lessons amidst the ongoing COVID-19 pandemic. The present findings indicate the need to develop an appropriate program to help students maintain a good QOL in the "new normal" era. Based on the results of this research, schools should pay attention to improving students' self-efficacy and qualities of social connection in stressful situations such as a pandemic. Students and educators should be aware of strategies that can enhance their self-efficacy and apply them in their attitudes and everyday pursuits.

## Strengths and limitations

The sample used in this study was drawn from all universities and university colleges with a PHN study programme in Norway, and the students were all at the same stage of the programme. The Norwegian school system is fairly homogenous, so it is likely that the findings would be similar in other student populations at the same level of higher education. However, several limitations of this study should be considered when interpreting the results. The sample size was fairly small, with a 63% response rate. Further, we have no information about the students who did not participate in this study. We cannot assess whether participants and non-participants differed in any respect. This research on QOL was based on PHN students'

subjective perspectives and refers to individual internal judgments about perceptions of QOL. Since the QOL is a complex phenomenon, it may be altered by aspects that are not explored in this study. Further, the phenomenon involves the students describing their life satisfaction. However, the QOL is relative phenomenon that can be influenced by individual needs and expectations. Despite these limitations, the study fills a research gap when it comes to the scarcity of data on the QOL of university students after the lockdown ended and has allowed several recommendations to be made.

## Conclusion

The results from this study showed that self-efficacy, satisfaction with physical health, loneliness and civil status were all significantly correlated with QOL during the first month of the COVID-19 pandemic. The strongest predictive factors of QOL among PHN students during the pandemic were loneliness and self-efficacy. Our results provide insight into the QOL of PHN students, which managers in the higher education sector can use to outline specific coping strategies that can help students during a pandemic.

## Acknowledgments

We would like to thank all the study participants.

## Author Contributions

**Conceptualization:** Lisbeth Valla, Bente Sparboe-Nilsen.

**Formal analysis:** Milada Cvancarova Småstuen.

**Methodology:** Milada Cvancarova Småstuen.

**Writing – original draft:** Lisbeth Valla, Victoria Telle Hjellset.

**Writing – review & editing:** Victoria Telle Hjellset, Milada Cvancarova Småstuen, Bente Sparboe-Nilsen.

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
