## [Decision Letter · Decision Letter 0]

4 Nov 2022

PONE-D-22-16970Quality of life and associated factors among Norwegian public health nurse students during the first month of the COVID-19 pandemic: a cross-sectional studyPLOS ONE

Dear Dr. Valla,

Thank you for submitting your manuscript to PLOS ONE. After careful consideration, we feel that it has merit but does not fully meet PLOS ONE’s publication criteria as it currently stands. Therefore, we invite you to submit a revised version of the manuscript that addresses the points raised during the review process.

ACADEMIC EDITOR: I received comments and recommendations from two reviewers. Their recommendations are mixed. Please carefully consider their comments, revise and improve your manuscript.

We look forward to receiving your revised manuscript.

Kind regards,

Jianguo Wang, PhD

Academic Editor

PLOS ONE

Journal Requirements:

4. We note you have included a table to which you do not refer in the text of your manuscript. Please ensure that you refer to Table 2 in your text; if accepted, production will need this reference to link the reader to the Table.

Reviewers' comments:

Reviewer's Responses to Questions

**Comments to the Author**

1. Is the manuscript technically sound, and do the data support the conclusions?

Reviewer #1: Yes

Reviewer #2: No

2. Has the statistical analysis been performed appropriately and rigorously? 

Reviewer #1: Yes

Reviewer #2: No

3. Have the authors made all data underlying the findings in their manuscript fully available?

Reviewer #1: Yes

Reviewer #2: No

4. Is the manuscript presented in an intelligible fashion and written in standard English?

Reviewer #1: Yes

Reviewer #2: Yes

5. Review Comments to the Author

Reviewer #1: This is an interesting paper but there are some issues that must be addressed before it can be published:

1. The authors should have put more emphasis on what is different about the Norwegian students? And how the structure made by Norwegian students is different when compared to the results from other countries?

2. It may be helpful to identify the target audience for the paper, given the rather technical nature of research topic.

3. Identify the gap in exiting literature, by arguing what is missing or inadequate in existing solutions and thus your study is necessary. This needs to be briefly noted in Introduction, and then further elaborated in the Literature Review, with in-depth analysis and substantiation of citations. For example some literature (not comprehensive) must pay attention to:

[1] Does low self-esteem predict anxiety among Chinese college students?. Psychology Research and Behavior Management, 15, 1481.

[2] Gender differences in the relationship between self-esteem and depression among college students: a cross-lagged study from China. Journal of Research in Personality, 97, 104202.

[3] Influencing factors, prediction and prevention of depression in college students: A literature review. World Journal of Psychiatry, 12(7), 860.

4. Discussion of results/findings, needs to related to previous literature and compare and contrast the findings/claims against that of previous studies

Reviewer #2: This study investigated the association between quality of life (QOL) and civil status, self-efficacy, loneliness, and physical and mental health among public health nurse (PHN) students during the COVID-19 pandemic in April. However, the selected study design seems inadequate to the enough practical significance. The Methods section lacks additional details and standard procedures before an in-depth interpretation of the results can be made. Moreover, the study results were not presented comprehensively and clearly enough. In addition, the time of data collection (May to November 2020) did not coincide with the time of investigation for the purpose of the study (April 2020).

Introduction 1. The introduction should describe the particularities of the sample (PHN students) and the significance of the investigation. What is the significance of the study's selection of PHN students as subjects?

Methods

2. How do the authors determine the influencing factors of quality of life? According to published literature, the influencing factors included by the authors are insufficient. Physical and mental health, family life, social life, material conditions, living environment and other aspects may affect the quality of life.

3. The authors that the strongest predictive factors for QOL were found to be perceived loneliness and self-efficacy. However, this conclusion may be wrong due to the lack of inclusion of more important influencing factors.

4. The authors did not report the R-squared of the linear regression equation, so the contribution of the included factors to the quality of life cannot be judged.

Whether the recipient has specific requirements when answering the survey.

5. Inclusion criteria was not clearly described. The sample size should be reported based on statistical methods and calculation formulas.

6. Did the authors imply in the questionnaire that this was a survey based on the COVID-19 pandemic.

7. Additional details on the measure of quality of life are needed. Each of the items included in the Satisfaction with Life Scale should be described in detail, and the other scales should be similarly described.

8. “The model fit was good and all assumptions for linear regression were fulfilled.” The basis of this statement is not detailed.

9. Effect sizes (ES) and 95% confidence intervals should be reported.

Results

10. The details of research results of QOL and civil status, self-efficacy, loneliness, and physical and mental health had not reported and displayed.

11.Age in Table 2 should be included in the equation as a covariate

12. “Participants who were not married/living with a partner had QOL scores of around 0.5 points lower than participants living with a partner.” There is no basis for the presentation of this result.

Discussion 13. Line 219: The aim of the study was to evaluate the results in the first month of the COVID-19 pandemic, however, the statement in the method section was that the data were collected from May to November 2020. There is a contradiction between these two statements.

14.Lack of discussion on comparison of quality of life in normal period (non-pandemic period)

6. PLOS authors have the option to publish the peer review history of their article (what does this mean?). If published, this will include your full peer review and any attached files.

Reviewer #1: No

Reviewer #2: **Yes: **Huiming Huang

---

## [Author Response · Author response to Decision Letter 0]

16 Dec 2022

Response to rewievers

Journal Requirements:

Response: Thank you for your comments and for giving us the opportunity to review the manuscript. We have now, to the best of our ability complied the manuscript according to PLOS ONE's guidelines.

Response: We have now included in the Methods section on page 8 under the Ethics section a statement that a written informed consent was obtained from all participants upon recruitment and participants did not receive any financial compensation.

Response: The ORCID iD for the corresponding author is https://orcid.org/0000-0001-6742-7787. This is now validated in the Editorial Manager.

4. We note you have included a table to which you do not refer in the text of your manuscript. Please ensure that you refer to Table 2 in your text; if accepted, production will need this reference to link the reader to the Table.

Response: Thank you for pointing this out, we refer to Table 2 on p 9 in the revised manuscript. 

Review Comments to the Author

Reviewer #1: 

Comment 1: The authors should have put more emphasis on what is different about the Norwegian students? And how the structure made by Norwegian students is different when compared to the results from other countries?

Response 1: We appreciate your input, which will allow us to clarify this point.

Public health nurses’ education in Norway is a post-graduate program. One must be a licensed nurse with at least one year of work experience in order to be admitted to the program. The PHN education prepares nurses to become experts in health promotion and illness preventive work for children and adolescents aged 0 to 20 as well as their parents. Additionally, as part of their education, they have 10 weeks of practical experience in clinics and in school health services. Few other countries offer similar educational opportunities in health promotion and illness prevention work, although our graduates can be to some degree compared to students in other countries at a post graduated or higher University level. This student group stands out from other student groups because they are frequently older and thus have families and cohabitants and have previous experience with undergrad courses.

We have now added a few phrases to the manuscript on page 3 line 70-72 and page 4 line 103 -105.

Comment 2: It may be helpful to identify the target audience for the paper, given the rather technical nature of research topic.

Response 2: People who are engaged in higher education and thus deal with students, including college and university staff, are the intended audience for this study. We acknowledge the reviewers’ comments and have added a statement about the target audience for the paper on page 5 line 113 to 117 in response to the reviewers' suggestions. 

Comment 3: Identify the gap in exiting literature, by arguing what is missing or inadequate in existing solutions and thus your study is necessary. This needs to be briefly noted in Introduction, and then further elaborated in the Literature Review, with in-depth analysis and substantiation of citations. For example some literature (not comprehensive) must pay attention to:

[1] Does low self-esteem predict anxiety among Chinese college students?. Psychology Research and Behavior Management, 15, 1481.

[2] Gender differences in the relationship between self-esteem and depression among college students: a cross-lagged study from China. Journal of Research in Personality, 97, 104202.

[3] Influencing factors, prediction and prevention of depression in college students: A literature review. World Journal of Psychiatry, 12(7), 860.

Response 3:

We acknowledge the reviewers’ comments and have now briefly identified the gap in exiting literature in the Introduction section page 3 line 77 to 79. We have also included what is missing or is inadequate in existing research and thus why our study is necessary on page 4 line 106 to 110.

Comment 4. Discussion of results/findings, needs to related to previous literature and compare and contrast the findings/claims against that of previous studies

Response 4 To the best of our abilities, we have now discussed our findings in relation to the published literature and compared with the findings in the relevant literature.

Reviewer #2: 

This study investigated the association between quality of life (QOL) and civil status, self-efficacy, loneliness, and physical and mental health among public health nurse (PHN) students during the COVID-19 pandemic in April. However, the selected study design seems inadequate to the enough practical significance. The Methods section lacks additional details and standard procedures before an in-depth interpretation of the results can be made. Moreover, the study results were not presented comprehensively and clearly enough. In addition, the time of data collection (May to November 2020) did not coincide with the time of investigation for the purpose of the study (April 2020).

Comment 1: The introduction should describe the particularities of the sample (PHN students) and the significance of the investigation. What is the significance of the study's selection of PHN students as subjects?

Response 1: We considered our sample of PHN students’ representative for the PHN population. Further, despite this student population being older and perhaps more established – please see our reply above – compared to other student populations we feel that the results can be generalized to other graduate student population. Moreover, PHN graduates will be involved in public health discussions and Covid 19 or similar infections will be very relevant for them both as a health threat for themselves and a general public health issue.

Comment 2: Methods: How do the authors determine the influencing factors of quality of life? According to published literature, the influencing factors included by the authors are insufficient. Physical and mental health, family life, social life, material conditions, living environment and other aspects may affect the quality of life.

Response 2: We agree with the reviewer that quality of life is a complex issue and there are many variables which may possibly influence it. We have analyzed the variables which were available. We could have used a more lengthy questionnaire that would cover more variables however, the longer the questionnaire the more likely it is that the students do not want to participate or fill in the questionnaire properly. Thus, we acknowledge that we have analyzed only a selection of possible predictive factors, nevertheless we still consider our findings of interest. We have added to the Limitation section on page 14 line 325-326 that we are aware that our selection of possible predictive factors is limited. 

Comment 3: The authors that the strongest predictive factors for QOL were found to be perceived loneliness and self-efficacy. However, this conclusion may be wrong due to the lack of inclusion of more important influencing factors.

Response 3: We agree with the reviewer that there can always be more measured and unmeasured confounders. However, our aim was to investigate the available possible predictive factors and of those perceived loneliness and self-efficacy remained the strongest predictors. We have now added to the Limitation section that we are aware of unmeasured confounders. 

Comment 4 The authors did not report the R-squared of the linear regression equation, so the contribution of the included factors to the quality of life cannot be judged.

Whether the recipient has specific requirements when answering the survey.

Response 4 The R-square was 0.554 and the adjusted R-square was 0.546 thus the included covariates explained about half of the variation in the dependent variable. This information has been added to the results on page 10.

Comment 5. Inclusion criteria was not clearly described. The sample size should be reported based on statistical methods and calculation formulas.

Response 5 This study was not a randomized control trial but an observational, association study thus we did not estimate the needed sample size to reveal a given difference as statistically significant. As we intended to perform linear regression, we aimed at having at least 10-15 individuals per included covariate. With 275 participants we would have been able to fit a model with 25-27 covariates which was far more than necessary to answer our aim thus we consider our study sufficiently powered. 

Comment 6 Did the authors imply in the questionnaire that this was a survey based on the COVID-19 pandemic.

Response 6 We have indeed included a statement in the questionnaire that the survey was based on the covid 19 pandemic.

Comment 7 Additional details on the measure of quality of life are needed. Each of the items included in the Satisfaction with Life Scale should be described in detail, and the other scales should be similarly described.

Response 7 Thank you for your comment. We have now added all five items to the QOL form and the ten to the Self-Efficacy form at page 7. The students’ physical health was assessed using the question ‘How satisfied have you been with your physical health over the last four weeks? The students’ mental health was assessed using the question, ‘How satisfied have you been with your mental health, emotions and mood over the last four weeks? This is stated in the article page 7

Comment 8 “The model fit was good and all assumptions for linear regression were fulfilled.” The basis of this statement is not detailed.

Response 8 We have performed a visual inspection of the residuals (histograms, Q-Q plots) and considered the assumptions of linear regression to be fulfilled, e.g. the residuals were normally distributed with the standardized residuals having mean of 0 and SD=1.

Comment 9. Effect sizes (ES) and 95% confidence intervals should be reported.

Response 9 We completely agree with the reviewers and both the regression coefficients and effect sizes (ES) are included in table 2. 

Comment 10 Results. The details of research results of QOL and civil status, self-efficacy, loneliness, and physical and mental health had not reported and displayed.

Response 10: To the best of our abilities, we have now provided more details on research results concerning QOL and civil status, self-efficacy, loneliness, and physical and mental health in the Discussion section of the revised manuscript. We have also to a greater extend discussed our findings in relation to the published literature and compared and contrasted our findings/claims against those of previous studies.

Comment 11: Age in Table 2 should be included in the equation as a covariate

Response 11: Age in Table 2 is now included in the equation as a covariate.

Comment 12. “Participants who were not married/living with a partner had QOL scores of around 0.5 points lower than participants living with a partner.” There is no basis for the presentation of this result.

Response 12: We apologize for this mistaken and thank the reviewer for pointing this out. The regression coefficient is -0.17 and not -0.50. The interpretation should be as follows:

‘Participants who were not married/living with a partner had QOL scores of around 0.17 points lower than participants living with a partner.” 

Comment 13 Discussion, line 219: The aim of the study was to evaluate the results in the first month of the COVID-19 pandemic, however, the statement in the method section was that the data were collected from May to November 2020. There is a contradiction between these two statements.

Response 13 We agree with the reviewer that the aim was not formulated correctly regarding the time of data collection. It has been revised as follows: ‘The aim of the study was to evaluate the results of the first wave of the COVID-19 pandemic as the data were collected from May to November 2020.’

Comment 14 Lack of discussion on comparison of quality of life in normal period (non-pandemic period)

Response 14 We acknowledge the reviewers’ comments and have included a section and references in the Discussion on page 9 that our results were comparable to QOL in a nationwide survey of Norwegian students in the non-pandemic period where Norwegian students in higher education generally scored moderate to high on the SWLS Scale.

---

## [Decision Letter · Decision Letter 1]

19 Jan 2023

PONE-D-22-16970R1Quality of life and associated factors among Norwegian public health nurse students during the first months of the COVID-19 pandemic: a cross-sectional studyPLOS ONE

Dear Dr. Valla,

Thank you for submitting your manuscript to PLOS ONE. After careful consideration, we feel that it has merit but does not fully meet PLOS ONE’s publication criteria as it currently stands. Therefore, we invite you to submit a revised version of the manuscript that addresses the points raised during the review process.

We look forward to receiving your revised manuscript.

Kind regards,

Jianguo Wang, PhD

Academic Editor

PLOS ONE

Journal Requirements:

Additional Editor Comments:Please carefully address all commentsImprove the quality of the manuscript.

Reviewers' comments:

Reviewer's Responses to Questions

**Comments to the Author**

1. If the authors have adequately addressed your comments raised in a previous round of review and you feel that this manuscript is now acceptable for publication, you may indicate that here to bypass the “Comments to the Author” section, enter your conflict of interest statement in the “Confidential to Editor” section, and submit your "Accept" recommendation.

Reviewer #1: All comments have been addressed

Reviewer #2: (No Response)

2. Is the manuscript technically sound, and do the data support the conclusions?

Reviewer #1: Yes

Reviewer #2: Partly

3. Has the statistical analysis been performed appropriately and rigorously? 

Reviewer #1: Yes

Reviewer #2: Yes

4. Have the authors made all data underlying the findings in their manuscript fully available?

Reviewer #1: Yes

Reviewer #2: No

5. Is the manuscript presented in an intelligible fashion and written in standard English?

Reviewer #1: Yes

Reviewer #2: Yes

6. Review Comments to the Author

Reviewer #1: (No Response)

Reviewer #2: Most comments have been clarified and supplemented, however, in my opinion there are still several problems need to be improved.

1. The revised manuscript should be marked with different colors for reviewer.

2. Comment 2 and 3. In abstract section, the result of “The strongest predictive factors for QOL were found to be perceived loneliness and self-efficacy” should add “among these factors”.

3. Comment 5. Observational research also needs to describe the criteria for inclusion and deletion. Observational study should comply with the STROBE (Strengthening the Reporting of Observational Studies in Epidemiology) statement.

4. Comment 7. Additional protocol and reference on the measure of mental and physical health are needed.

5. Comment 10. Measurement results or questionnaire score of QOL and civil status, self-efficacy, loneliness, and physical and mental health should be reported and displayed. I suggest adding another table to present this results and change Table 2 to Table 3.

7. PLOS authors have the option to publish the peer review history of their article (what does this mean?). If published, this will include your full peer review and any attached files.

Reviewer #1: No

Reviewer #2: **Yes: **Huiming Huang

---

## [Author Response · Author response to Decision Letter 1]

14 Feb 2023

Reviewer #1: (No Response)

Reviewer #2: Most comments have been clarified and supplemented, however, in my opinion there are still several problems need to be improved.

Comment 1: The revised manuscript should be marked with different colors for reviewer.

Response 1: Thank you for giving us the opportunity to review this manuscript. Now, all changes made in both the previous revision and the present revision are highlighted in red.

Comment 2: In abstract section, the result of “The strongest predictive factors for QOL were found to be perceived loneliness and self-efficacy” should add “among these factors”. 

Response 2: This is now included in the abstract page 2 line 42.

Comment 3: Observational research also needs to describe the criteria for inclusion and deletion. Observational study should comply with the STROBE (Strengthening the Reporting of Observational Studies in Epidemiology) statement

Response 3: We acknowledge the reviewer’s comments and have now clarified the criteria for inclusion in the study on page 5 and 6 line 129-136.

All students, both full-time and part-time, enrolled in PHN educational programme in eight Universities and Colleges in Norway during the COVID-19 pandemic met the inclusion criteria for this study. The participating universities were Oslo Metropolitan University, VID Specialized University, University of South-Eastern Norway (USN), the Norwegian University of Science and Technology (NTNU), University of Stavanger (UIS), The Arctic University of Norway (UIT), Inland Norway University of Applied Sciences and The Western Norway University of applied Sciences.

Comment 4: Additional protocol and reference on the measure of mental and physical health are needed.

Response 4: 

We acknowledge the reviewer’s comments and references on the measures about mental and physical health are now included in the method section page 7 and 8. 

The questions about mental and physical health are from the National Norwegian Recommendations for measuring subjective quality of life in Norway. The recommendations are based on a review of research and international recommendations, input from national and international actors in the field, as well as professional assessments of quality of life. (Nes, Hansen and Barstad, 2018). The Norwegian recommendations are based on the Dolan and Metcalf’ Recommendations on measures for use by national governments, which recommend that both physical and mental health be included as questions in measures of cognitive subjective quality of life. (Dolan and Metcalf, 2012). Both questions about mental and physical health have been tested and validated in a Norwegian setting (Berg, Berglund and Lund, (2018).

Comment 5: Measurement results or questionnaire score of QOL and civil status, self-efficacy, loneliness, and physical and mental health should be reported and displayed. I suggest adding another table to present these results and change Table 2 to Table 3.

Response 5: We agree with the reviewer’s comments. We have now added a new table with descriptive statistics on the questionnaire score of QOL, civil status, self-efficacy, loneliness, and physical and mental health. As the responses on physical and mental quality of life are continuous variables, the data are presented as median and ranges. The results are now included in table 2 on page 10.

---

## [Decision Letter · Decision Letter 2]

21 Feb 2023

Quality of life and associated factors among Norwegian public health nurse students during the first months of the COVID-19 pandemic: a cross-sectional study

PONE-D-22-16970R2

Dear Dr. Valla,

We’re pleased to inform you that your manuscript has been judged scientifically suitable for publication and will be formally accepted for publication once it meets all outstanding technical requirements.

Kind regards,

Jianguo Wang, PhD

Academic Editor

PLOS ONE

Additional Editor Comments (optional):

Reviewers' comments:

Reviewer's Responses to Questions

**Comments to the Author**

1. If the authors have adequately addressed your comments raised in a previous round of review and you feel that this manuscript is now acceptable for publication, you may indicate that here to bypass the “Comments to the Author” section, enter your conflict of interest statement in the “Confidential to Editor” section, and submit your "Accept" recommendation.

Reviewer #1: All comments have been addressed

Reviewer #2: All comments have been addressed

2. Is the manuscript technically sound, and do the data support the conclusions?

Reviewer #1: Yes

Reviewer #2: Yes

3. Has the statistical analysis been performed appropriately and rigorously? 

Reviewer #1: Yes

Reviewer #2: Yes

4. Have the authors made all data underlying the findings in their manuscript fully available?

Reviewer #1: Yes

Reviewer #2: Yes

5. Is the manuscript presented in an intelligible fashion and written in standard English?

Reviewer #1: Yes

Reviewer #2: (No Response)

6. Review Comments to the Author

Reviewer #1: I recommend acceptance. The author has made changes in response to comments and I have no further comments.

Reviewer #2: (No Response)

7. PLOS authors have the option to publish the peer review history of their article (what does this mean?). If published, this will include your full peer review and any attached files.

Reviewer #1: No

Reviewer #2: **Yes: **Huiming Huang

---

## [Editor Report · Acceptance letter]

6 Mar 2023

PONE-D-22-16970R2 

Quality of life and associated factors among Norwegian public health nurse students during the first months of the COVID-19 pandemic: a cross-sectional study 

Dear Dr. Valla:

I'm pleased to inform you that your manuscript has been deemed suitable for publication in PLOS ONE. Congratulations! Your manuscript is now with our production department. 

Kind regards, 

on behalf of

Dr. Jianguo Wang 

Academic Editor

PLOS ONE